# Clinical Significance of Elevated KSHV Viral Load in HIV-Related Kaposi’s Sarcoma Patients in South Africa

**DOI:** 10.3390/v16020189

**Published:** 2024-01-26

**Authors:** Rebecca Monica Tibenderana, Melissa Jayne Blumenthal, Emmanuel Bukajumbe, Georgia Schäfer, Zainab Mohamed

**Affiliations:** 1Department of Radiation Oncology, University of Cape Town, Cape Town 7925, South Africa; beckytibenderana@yahoo.com; 2Department of Integrative Biomedical Sciences, Institute of Infectious Disease and Molecular Medicine, University of Cape Town, Cape Town 7925, South Africa; georgia.schafer@icgeb.org; 3International Centre for Genetic Engineering and Biotechnology (ICGEB), Cape Town 7925, South Africa; 4Aberdeen Surgery, Aberdeen, NSW 2336, Australia; ebukajumbe@gmail.com; 5Hatchile Consult Ltd., Kampala 759125, Uganda

**Keywords:** Kaposi’s sarcoma, KSHV, HHV-8, HIV, viral load, AIDS-related malignancy

## Abstract

Kaposi’s sarcoma (KS) is an AIDS-defining illness caused by Kaposi’s sarcoma-associated herpesvirus (KSHV) predominantly in the context of HIV-related immune suppression. We aimed to explore the usefulness of KSHV DNA viral load (VL) measurement in predicting the severity, response to treatment and outcome of KS. We retrospectively assessed a cohort of KS patients (*n* = 94) receiving treatment at Groote Schuur Hospital, Cape Town, South Africa. Demographic and clinical data, KS staging and response to treatment were extracted from patient files, while long-term survival was ascertained from hospital records. KSHV serology and VL and hIL-6 were determined empirically from patients’ blood. All patients were HIV-positive adults, the majority of whom were on HAART at the time of recruitment. KSHV VL was detectable in 65 patients’ blood (median: 280.5/10^6^ cells (IQR: 69.7–1727.3)) and was highest in patients with S1 HIV-related systemic disease (median 1066.9/10^6^ cells, IQR: 70.5–11,269.6). KSHV VL was associated with the S1 stage in a binomial regression controlling for confounders (adjusted odds ratio 5.55, 95% CI: 1.28–24.14, *p* = 0.022). A subset of six patients identified to have extremely high KSHV VLs was predominantly T_1_ stage with pulmonary KS, and most had died at follow-up. In our cohort, elevated KSHV VL is associated with systemic HIV-related illness in KS disease. Extremely high KSHV VLs warrant further investigation for patients potentially requiring intensive treatment and investigation for progression or diagnosis of concurrent KSHV lytic syndromes.

## 1. Introduction

Kaposi’s sarcoma (KS) was the first malignancy to be classified as an Acquired Immune Deficiency Syndrome (AIDS)-defining illness in 1981 and is the commonest AIDS-related malignancy in sub-Saharan Africa [1,2,3,4]. KS is a multicentric, highly vascularised [1] tumour comprised of hyperproliferating spindle cells and infiltrating monocytes, T-cells and plasma cells [5,6]. Clinically, KS commonly presents as a multifocal, flat, red/purple patch of variable size on the skin and progresses to cutaneous plaques and nodules usually on the head, neck or lower limbs [6,7]. Alternative presentation sites are the oral mucosa, the lungs, gastrointestinal tract or the lymph nodes, often accompanied with marked lower limb lymphoedema [7,8]. Four types of KS have been described, namely AIDS-related, classic, iatrogenic or transplant-associated and endemic. The gammaherpesvirus, Kaposi’s sarcoma-associated herpesvirus (KSHV), is the causative agent of all types of KS, but while necessary, it is not sufficient for KS development, requiring immune suppression. In the sub-Saharan African setting, this is often, although not always, due to HIV infection (AIDS-related KS). Both HIV infection and KSHV lytic infection promote cytokine production, which in turn may initiate KS tumour development and progression. In addition to KS, KSHV is the etiological agent of primary effusion lymphoma (PEL), multicentric Castleman disease (MCD) and KSHV-inflammatory cytokine syndrome (KICS), which also often occur in HIV-positive immunosuppressed patients [9,10,11,12,13,14,15]. These KSHV-related pathologies, which often overlap, pose diagnostic challenges in resource-limited settings [14,15].

The diagnosis of KS Is made by clinical evaluation and confirmed with positive LANA1 staining histology, which usually shows proliferation of abnormal vascular spaces and spindle cells. Other investigations to define the extent of disease include chest radiograph, stool occult blood, CD4 count and HIV viral load (VL) [16]. Staging of KS utilizes the AIDS Clinical Trials Group (ACTG) criteria formulated by Krown et al. [17,18] and adapted following the introduction of ART [19,20,21]. ACTG staging of KS refers to tumour extent (T), immunological status as defined by the CD4 count (I) and presence of systemic illness (S), with the classification of good risk (subscript 0) or poor risk (subscript 1) [17,18,20].

The mainstay of treatment for HIV-associated KS is HAART [22]. Localised KS may be additionally treated with radiotherapy, cryotherapy or intralesional vinca alkaloids. In sub-Saharan Africa, most patients with HIV-associated KS present with advanced disease, including KS-associated lymphoedema and pulmonary and gastrointestinal involvement, and systemic treatment is indicated to improve quality of life [23,24]. At Groote Schuur Hospital, patients are treated with intravenous bleomycin at 10 IU/m^2^ and vincristine at 1.4 mg/m^2^ in addition to HAART as the first-line therapy administered every two weeks for 6–20 cycles, depending on tolerability and clinical response.

It has been suggested that KSHV viraemia may be an indicator for the clinical severity of KS [7,25]. KSHV VL levels in peripheral blood differ between KSHV-associated diseases, with the highest levels evident in lytically associated syndromes such as MCD and KICS, and lower, although still elevated levels, in PEL and KS [11,26,27]. Furthermore, KSHV VL in blood and oral fluids has been shown to be associated with KS disease status (progressing, stable and regressing) [28,29] and severity [9], and KSHV lytic reactivity during KS [30] has been proposed as a clinical tool for assessment of the risk of KS progression [9,26,27,28,29,30]. KSHV VL has been characterised as a virological parameter, together with viral interleukin 6 (vIL-6), to monitor KSHV-MCD treatment progress and outcome in combination with immunological parameters, such as C-reactive protein (CRP), haemoglobin, albumin, sodium, platelets and host interleukin 6 (IL-6), to characterise a profile corresponding to best clinical response [26,31,32]. KSHV VL elevated above 100 copies/10^6^ cells is included in the working case definition of KICS [9,10], and we have previously demonstrated that elevated KSHV VL has prognostic utility in that it was associated with increased mortality in both a cohort of hospitalized HIV-infected patients and, similarly, in a cohort of hospitalized COVID-19 patients in Cape Town, South Africa [33,34]. In a cohort of KS patients with mostly T_1_ disease in Harare, Zimbabwe, pre-treatment plasma KSHV levels of less than 660 copies/mL were associated with improved survival and with a better clinical response [35].

KSHV VL has been proposed as a clinical tool for assessment of risk of KS progression [10,26,27,28,29,30] and may be a useful biomarker for KSHV-associated disease risk, to inform diagnosis and to manage KSHV-associated diseases. This study aimed to further assess the applicability of KSHV VL as a surveillance tool for KS disease severity, treatment response and outcome in a high-HIV setting with the overall goal of drawing clinicians’ and policymakers’ attention to the importance of KSHV diagnostics in clinical care.

## 2. Materials and Methods

### 2.1. Study Participants

In this retrospective cohort study, we reviewed medical records and laboratory results of 100 patients with symptomatic KS who received treatment for KS at Groote Schuur Hospital, Department of Radiation Oncology, between 2014 and 2018, and were recruited in a related study [36]. Groote Schuur Hospital is a public healthcare facility servicing the greater Western Cape area via a referral system. Eligible participants were male or female, over 18 years of age and had a diagnosis of KS either clinically or histologically confirmed. All patients had a thorough clinical examination by experienced clinicians to document the occurrence of typical KS cutaneous lesions, mucosal lesions and lymphoedema. When indicated, skin biopsy and chest X-ray supported the diagnosis of KS. Patients with classic or endemic KS and those with missing medical records were excluded. Of the 100 participants, 2 were HIV-negative, while clinical records of 4 patients could not be traced; these 6 patients were therefore excluded from this study. Demographic and clinical information collected from medical records of the remaining *n* = 94 participants included predominant site of disease, presence of HIV-related systemic illness, chemotherapy received, HAART use and response to treatment. Laboratory results including haematology, chemistry, CD4 count and HIV VL were obtained from the National Health Laboratory Service (NHLS) system, and radiology data were obtained from the hospital’s radiology archives. Patients were recruited regardless of when they received treatment, that is, either pre-chemotherapy at the first oncology visit, mid-chemotherapy (defined as having received at least four cycles of chemotherapy at the time of sample collection) or post-chemotherapy (defined as the end point of treatment for the patient), although the timing of treatment was recorded for consideration in statistical analysis.

Patients were staged according to the extent of their tumour (T), severity of immune suppression (I) and presence of other HIV-related systemic illness (S) and classified as good risk (subscript 0) or poor risk (subscript 1), according to the ACTG KS staging criteria [17,18,20]. Tumour extent was defined as poor risk (T_1_) if there was evidence of extensive oral cavity involvement, lymphoedema, tumour-related ulceration or visceral disease. Severe immune suppression (I_1_) was defined as CD4 count <200 cells/µL. Systemic symptoms were defined as poor risk (S_1_) if the patient had a history of any opportunistic infections, presence of B symptoms (namely ≥10% weight loss, unexplained fevers or night sweats or persistent diarrhea), another HIV-related illness or a Karnofsky performance status score <70. The clinical extent of T_1_-stage KS was further subdivided based on predominant site of KS into lymphoedema, extensive cutaneous disease, pulmonary KS and gastrointestinal tract (GIT) or other visceral sites. KS immune reconstitution inflammatory syndrome (IRIS) was not applicable in this cohort. Although there were some patients with disease in more than one site, the predominant site was used for assessment of a patient’s T_1_ substage.

We defined the patients’ response to first-line treatment within a year of completion of chemotherapy as follows: complete response (resolution of macroscopically visible KS lesions); partial response (i.e., a 50% reduction in number or size of KS lesions); progressive disease (i.e., at least 25% increase in number or size of KS lesions); relapse (i.e., recurrence of symptomatic KS after completing systemic chemotherapy) [23,37]. HAART defaulters were defined as patients who had stopped antiretroviral therapy for more than 30 days [38]. Hospital and laboratory records were accessed in 2023 to ascertain long-term outcome. Duration of follow-up ranged from 64 to 104 months with mean follow-up of 84 months (7 years).

### 2.2. KSHV Serology and Quantification and hIL-6 Assays

Peripheral blood samples were drawn from patients during their routine visit. The plasma was assessed for anti-KSHV antibodies by ELISAs specific for the latency-associated nuclear antigen (LANA, or ORF73) and the lytic structural glycoprotein K8.1, using previously described methods [39]. KSHV DNA was detected in DNA extracted from whole blood samples with plasma previously removed using previously reported primers for the K6 gene region and an FAM/TAMRA-labelled probe [40] and quantified against a standard curve of DNA from plasmid-containing K6, as previously described [41,42]. Plasma human (h)IL-6 was measured using the Human IL-6 SimpleStep ELISA kit (Abcam), with a minimum detectable dose of 1.6 pg/mL quantified against a standard curve of hIL-6 recombinant protein. The reference median for IL-6 in HIV-infected patients is 1.80 pg/mL (interquartile range, 1.20–2.89 pg/mL) [43].

### 2.3. Statistical Analysis

Statistical analysis was performed in SPSS version 25 (IBM Corp, Armonk, NY, USA, 2017). Graphical representations were generated in Prism (v10; GraphPad Software Inc., San Diego, CA, USA). Univariate analyses consisted of non-parametric Wilcoxon rank-sum tests (for comparing two independent groups) or Kruskal–Wallis tests (for comparing more than two independent groups) and Fisher exact tests, as appropriate and indicated in the figure or table legends. Multivariate analyses were performed using binomial logistic regression for the categorical dependent variable, “S1 stage”, in relation to the specified covariates. Covariates were selected in an attempt to control for confounding factors as much as possible in our clinical cohort. Linearity of the continuous variables with respect to the logit of the dependent variable was confirmed via the Box–Tidwell procedure [44], and studentized residuals with values <2.5 standard deviations were accepted. KSHV VL was log-transformed to make the units more biologically understandable. *p* values are 2-tailed and were considered statistically significant and indicated in the figures and tables if <0.05. Participants with missing data were excluded pairwise in each analysis. Statistical outliers were identified by the Tukey method.

### 2.4. Ethics

All protocols were approved by the Human Research Ethics Committee, Health Sciences Faculty, University of Cape Town (Approval HREC/REF: 773/2020 and 279/2008) and were in accordance with the ethical standards of the 1964 Helsinki Declaration and its later amendments. Written informed consent was obtained from all individual participants in the study.

## 3. Results

### 3.1. Patient Characteristics

All patients presented to the Groote Schuur Radiation Oncology Department for diagnosis and/or treatment of confirmed KS (*n* = 100). Two HIV-negative patients and four patients whose records could not be traced were excluded from this analysis. Demographic and clinical information for the patients included in this analysis (*n* = 94) is presented in Table 1. The patients were predominantly male (63%) with a median age of 37 years (IQR: 32–42). While the majority (97%) of patients were on ART at recruitment, a large proportion of the patient cohort had a history of ART default (16%) or their default history was unknown (82%), and only 9% of patients had been on ART for at least a year (Table 1). Median CD4 count in the cohort at the time of diagnosis was 186 cells/µL (IQR: 55–341, Table 1 and Figure 1A), and median HIV VL was 49 copies/mL (IQR: 1–5553, Table 1 and Figure 1B). Despite HAART, 49% of patients were classified as I_1_ (immune status: poor risk) based on CD4 counts at diagnosis <200 cells/µL (Table 1 and Figure 1A). Most of the patients had T_1_ (tumour status: poor risk) disease at the time of presentation to the clinic (97%), with the majority presenting with lymphedema as the predominant site of disease (57%) followed by extensive cutaneous disease (21%), pulmonary involvement (12%) and GIT or other visceral site involvement (7%). Some patients had more than one site of disease, but we categorized them based on the predominant site. Few patients (10%) presented with S1 (systemic illness status: poor risk) KS.

In general, patients were anaemic, the median haemoglobin amongst females was 10.1 g/dL (IQR: 9.0–11.4) and males 11.2 g/dL (IQR: 9.8–13.1, Table 1 and Figure 1D), and they had low albumin with a median of 34 g/L (IQR: 26–40, Table 1). The median white cell count was 6.7 × 10^9^ cells/L (IQR: 5.0–8.8, Table 1 and Figure 1E), and the platelet count was 320.5 × 10^9^ cells/L (232.0–413.0, Table 1). Generally, patients had raised hIL-6 levels with a median of 21.3 pg/mL (9.9–4.4, Table 1 and Figure 1C). KSHV serology by ELISA to KSHV lytic antigen K8.1 and latent antigen LANA showed 99% serum positivity with 98% of the cohort positive for K8.1 and 71% positive for LANA (Figure 2A,B). KSHV VL was detectable in 68% of the cohort and was found to be elevated (>100 viral copies/10^6^ cells) in 49%. The median KSHV VL was 280.5 copies/10^6^ cells (IQR: 69.7–1727.3) with six patients displaying strikingly high KSHV VLs (Figure 2C). These six patients’ KSHV VLs were statistically classified as extreme outliers as they were greater than the 90th percentile of the data set. KSHV VL was not significantly correlated with CD4 count or HIV VL.

Following completion of chemotherapy (within 1 year of diagnosis), 42% of patients showed a response to chemotherapy, although only 3% had a complete response while 39% had a partial response. In total, 13% of patients did not respond to chemotherapy (2% had disease progression and 11% had disease relapse). We were unable to ascertain the response to treatment of 44% of the cohort due to loss to follow-up (LTFU, 23%) or death (21%). Long-term follow-up (mean follow-up time of 7 years since recruitment) revealed that 36 patients had died (38%, mean time to death of 16.6 months), 36 patients were still living (38%) and 22 patients were LTFU (24%, Table 1).

### 3.2. Association of KSHV VL with KS Severity, Response to Treatment and Outcome

None of the staging categorizations of KS severity had statistically significant higher median KSHV VLs (Figure 3) nor did any grouping show significantly different proportions of KSHV VL detectability in the blood. Patients with extremely high KSHV VLs statistically determined to be outliers (identified in Figure 2C and labelled in Figure 3A–D) were noted to be all categorized as stage T_1_ (Figure 3A) with a predominant site of disease (T substage) of pulmonary or GIT involvement (Figure 3B). The majority were staged as I_1_ (Figure 3C). Four of the six outliers were categorized as S_0_ stage and two stage S1. Even so, median KSHV VL was higher in the S1 category (median = 1066.9/10^6^ cells, IQR: 70.5–11,269.6) compared to the S_0_ category (median = 272.9/10^6^ cells, IQR: 79.95–1311.9, Figure 3D). Moreover, in a multivariate logistic regression controlling for sex, age, treatment stage at time of sample collection, CD4 count and hIL-6 levels, KSHV VL was statistically significantly associated with S1 stage (OR 5.55 (95% CI: 1.28–24.14), *p* = 0.022, Table 2).

Response to treatment following the completion of first-line chemotherapy was recorded as complete response (3 patients (3%)), partial response (36 patients (38%)), progressive disease (2 patients (2%)), relapse (10 patients (11%)) and death, which were the outcomes of 20 patients (21%). Twenty-three patients were lost to follow-up (24%). KSHV VL was not associated with response to treatment (Figure 4), although three from the subset of six patients with extremely high VLs (statistical outliers identified in Figure 2C) had an unknown response to treatment (Figure 4); therefore, we do not know how their response to treatment would have influenced the analysis. Of the other three patients with outlier KSHV VLs, one died, and two had a partial response to chemotherapy (Figure 4).

KSHV VL was not associated with long-term outcome (Figure 5), although as with response to treatment, it is unclear how the unknown survival status of patients who were LTFU would affect this result. Of the patients with outlier KSHV VLs, four died, one was LTFU and one was still living (Figure 5).

### 3.3. Severe Clinical Features of Patients with Extremely Elevated KSHV VL

The six patients with extremely high KSHV VLs classified as statistical outliers from the distribution of KSHV VLs in the patient cohort (Figure 2C) had features of severe KS disease. All of these patients presented with advanced tumour extent (staged as T_1_, Figure 3A and Table 3) with predominant site of disease noted as pulmonary (67%) or GIT/GIT/another visceral organ KS (33%). While T_1_ presentation was general to the entire cohort, the proportion of patients within this subset presenting with pulmonary KS or KS in the GIT/other visceral site was significantly higher than in the remainder of the cohort (67% vs. 8%, *p* < 0.001 and 33% vs. 6%, *p* = 0.014, respectively, Table 3). The majority of this subset of patients had an unknown response to treatment due to LTFU but at long-term follow-up were found to have died. All six patients were anaemic compared to a 76% prevalence of anaemia in the remainder of the cohort, and these patients had lower median CD4 counts compared to the remainder of the cohort. However, these were not statistically significant (Table 3). The KSHV VL outliers had significantly higher levels of serum IL-6 compared to the remainder of the cohort (*p* = 0.040, Table 3).

## 4. Discussion

Despite the introduction of HAART being widely considered the first line of treatment for KS, KS still poses a major challenge in the South African clinical context, with the majority of KS cases presenting with advanced disease resulting in poor response to treatment and outcomes. KSHV VL in the blood is not routinely assayed clinically but has been proposed as a surveillance biomarker for KS disease severity and progression as well as an indicator of other KSHV-associated diseases that pose diagnostic challenges in resource-limited settings [29,30]. We aimed to assess KSHV VL levels in a cohort of KS patients in South Africa to further inform the applicability of VL as a surveillance tool for KS disease severity, treatment response and outcome in a high-HIV setting.

Patients with a positive diagnosis of KS were recruited to our study. The vast majority of these patients were HIV-positive (98%), corroborating epidemiologists’ suggestions that HIV-associated KS accounts for the major KS burden in South Africa. The two patients with HIV-negative KS were excluded from this analysis because they did not fit the inclusion criteria. All but one patient in our cohort was on ART, but a significant proportion of patients defaulted, or their adherence was unknown, and only a small percentage of patients (9%) had been on ART for more than a year prior to KS diagnosis. Among our cohort, median CD4 counts at KS diagnosis (186, IQR: 55–341) and HIV VLs (49, IQR: 1–5553) indicate that KS diagnoses are not made exclusively in patients with uncontrolled HIV- and AIDS-related immune suppression but also in patients with controlled HIV infection. That said, there is a significant proportion of our cohort who defaulted from ART treatment, or this was not clear from the medical records, highlighting ART compliance as a likely contributing factor to KS diagnoses.

In this cohort of HIV-positive KS patients, KSHV VL was associated with S1-stage KS when controlling for sex, age, treatment stage at time of recruitment and CD4 count. A limitation of this analysis is that the percentage of patients with S1-stage KS was low (10%) compared to those with S_0_ stage; there was an imbalance in the outcome groups of our logistic regression. We therefore interpret this result with caution. Patients staged as S1 have evidence of systemic illness presenting as B symptoms, opportunistic infections, HIV-related illness or a Karnofsky performance status score < 70. The association between elevated KSHV VL and S1 staging could mean either that systemic illness causes or allows for reactivation of KSHV in the blood or that highly elevated KSHV in the blood contributes to the progression of systemic disease. Our current study design cannot differentiate between these, and as mentioned our sample size of S1-stage patients is limiting, but this result highlights an important consideration for future research.

KSHV VL has been suggested to be a useful biomarker that could be employed clinically to differentiate between KSHV-associated pathologies (that can be concurrent with KS) and KS disease progression [22]. Lytic-associated KSHV-related pathologies, namely MCD and KICS, are characterised by highly elevated KSHV VL in the blood, whereas KS and PEL patients generally have lower, although still possibly elevated, KSHV VL levels [22,23,41]. The KSHV VLs seen in our study, particularly that of the six outliers with extremely high KSHV VLs, more similarly match levels seen in MCD or KICS patients than would be expected in KS patients. It is known that diagnoses of other KSHV-related pathologies concurrent with KS can be technically difficult, especially in resource-limited settings. In our cohort, further investigation was not clinically indicated in these patients, but we cannot rule out the presence of other concurrent KSHV-related diseases. KSHV VL has been shown to be associated with KS disease status (progressing, stable and regressing) [24,25] and proposed as a clinical tool for assessment of KS progression [9,22,23,24,25,26]. A limitation of our study is that samples were not collected longitudinally, and so we cannot comment on the dynamics of KSHV VL through the progression of KS. As a non-randomized retrospective study, our study has a high risk of selection bias and unmeasured confounding.

Elevated KSHV VL has previously been shown to have prognostic utility, in that it was associated with mortality in a cohort of hospitalized HIV-infected patients and, similarly, in a cohort of hospitalized COVID-19 patients in Cape Town, South Africa [29,30]. In this cohort of KS patients, we did not see an association with long-term outcome nor with treatment response. However, we did note that the majority of the patients with extremely high KSHV VLs died.

## 5. Conclusions

In our study, KSHV VL in the blood is associated with systemic HIV-related illness (S1 staging) in KS disease. While our study design cannot differentiate between KSHV causing disease progression and severely progressed disease causing KSHV lytic activation, this relationship prompts further investigation into the biological mechanisms of the interplay between KSHV and HIV. A subset of KS patients with extremely high KSHV VLs and severe clinical features highlights KSHV VL as a putative biomarker in resource-limited, high-HIV settings, identifying patients who require intensive treatment and further investigation for KS progression or diagnosis of other concurrent KSHV lytic syndromes.

## Figures and Tables

**Figure 1 viruses-16-00189-f001:**
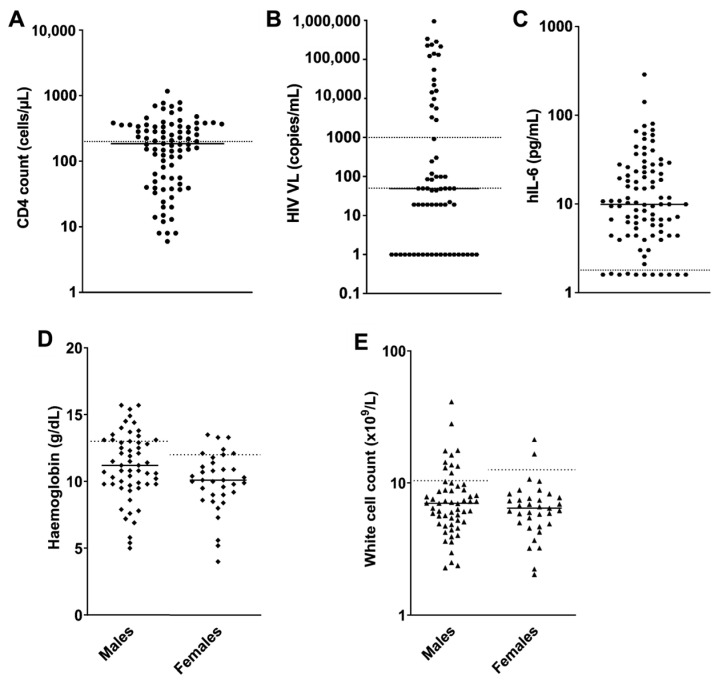
Scatter plots indicating selected laboratory abnormalities among KS patients (*n* = 94). (**A**) CD4 count at time of KS diagnosis. Dotted line indicates CD4 = 200 cells/µL, which is the cutoff for immune suppression staging (I) of KS, and solid line indicates median value; (**B**) HIV VL. Dotted lines indicate HIV VL = 50 copies/mL below, which is considered viral suppression, and 1000 copies/mL above, which is considered viremic. Patients with HIV VLs between 50 and 1000 copies/mL are considered to have low-level viremia. Solid line indicates median. (**C**) hIL-6 plasma levels. Dotted line indicates hIL-6 = 1.6 pg/mL, which is the reference value for HIV-positive patients. Solid line indicates median. (**D**) Haemoglobin levels. Dotted lines indicate the minimum value in the normal range for males (13 g/dL) and females (12 g/dL), and solid lines indicate median value; and (**E**) white cell count. Dotted lines indicate the minimum value in the normal range for males (10.4 × 10^9^/L) and females (12.6 × 10^9^/L), and solid lines indicate median value.

**Figure 2 viruses-16-00189-f002:**
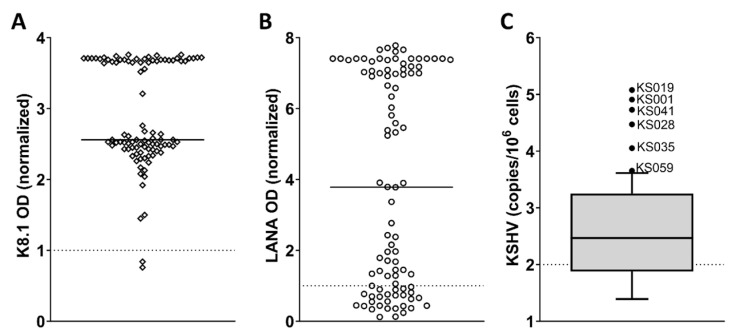
KSHV serology and viral load among KS patients (*n* = 94). (**A**) K8.1 ELISA optical density (OD) values measured in patient serum. Dotted line indicates the cut-off for a positive assay, and solid line indicates the median value; (**B**) LANA ELISA optical density values measured in patient serum. Dotted line indicates the cut-off for a positive assay, and solid line indicates the median value.; (**C**) KSHV viral load (log-transformed) at time of recruitment. Box and whiskers and outliers are determined by the Tukey method. Outliers are indicated with a dot and labelled. Dotted line indicates the cut-off for elevated KSHV viral load (100 copies/10^6^ cells), and solid line indicates the median value.

**Figure 3 viruses-16-00189-f003:**
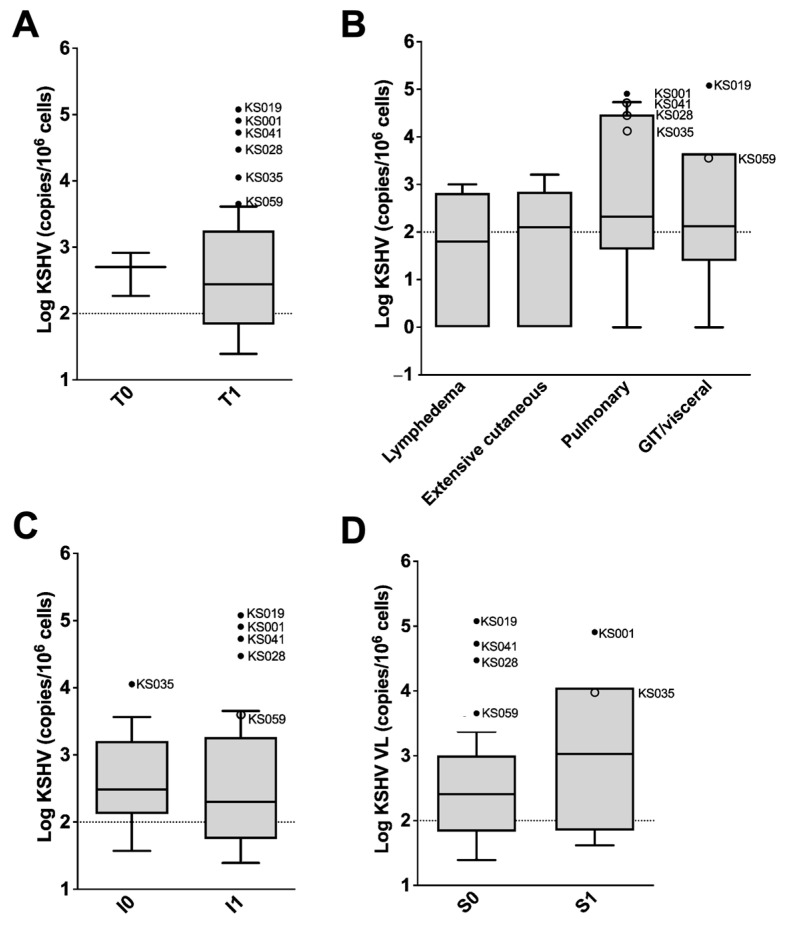
KSHV viral load by KS staging at diagnosis as an indication of severity among KS patients (*n* = 94). (**A**) T staging representing tumour extent; (**B**) T_1_ substaging based on site of predominant disease at diagnosis; (**C**) I staging representing immune suppression; and (**D**) S staging representing systemic disease. KSHV viral load outliers identified in Figure 2C are indicated with dots in these graphs (solid dots indicate outliers in the distributions here as well as in Figure 2C, whereas empty dots indicate that the values are not considered outliers in this distribution). Box and whiskers are determined by the Tukey method. The dotted lines indicate the cut-off (100 copies/10^6^ cells) for elevated KSHV viral load. Solid lines indicate median. Medians are compared statistically using the Wilcoxon rank-sum test (**A**,**C**,**D**) or Kruskal–Wallis test (**B**); *p* values > 0.05 are not shown. Samples without detectable KSHV DNA are excluded.

**Figure 4 viruses-16-00189-f004:**
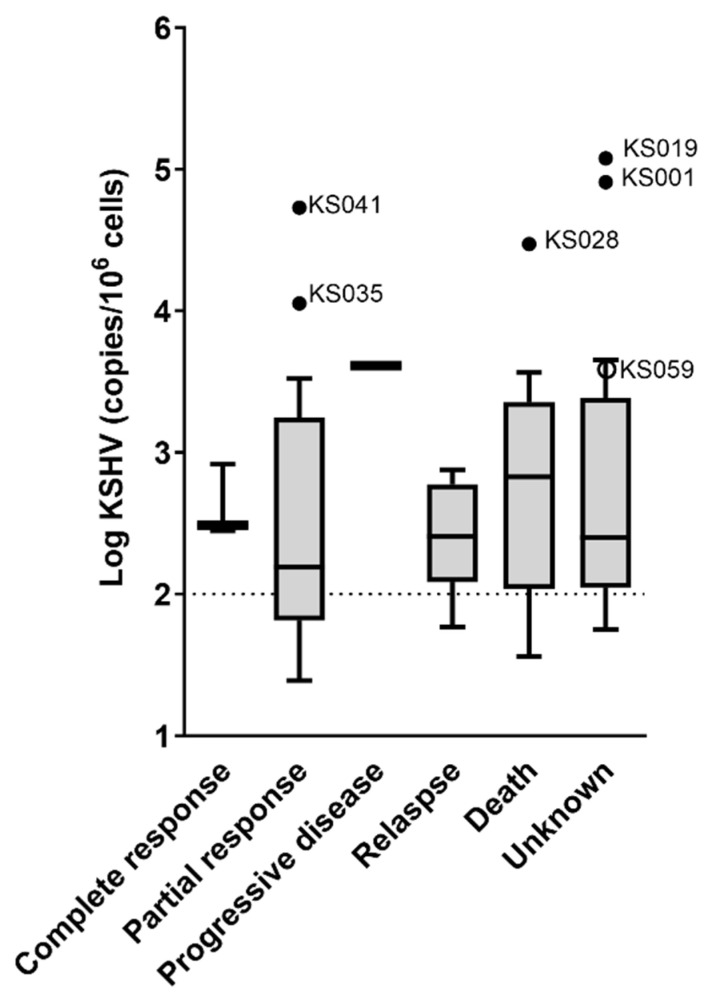
The association of KSHV VL to response to treatment among KS patients (*n* = 94). KSHV viral load outliers identified in Figure 2C are indicated with dots in these graphs (solid dots indicate outliers in the distributions here as well as in Figure 2C, whereas empty dots indicate that the values are not considered outliers in this distribution). The dotted lines indicate the cut-off (100 copies/10^6^ cells) for elevated KSHV viral load. Medians are compared statistically using the Kruskal–Wallis test; *p* values > 0.05 are not shown. Samples without detectable KSHV DNA are excluded.

**Figure 5 viruses-16-00189-f005:**
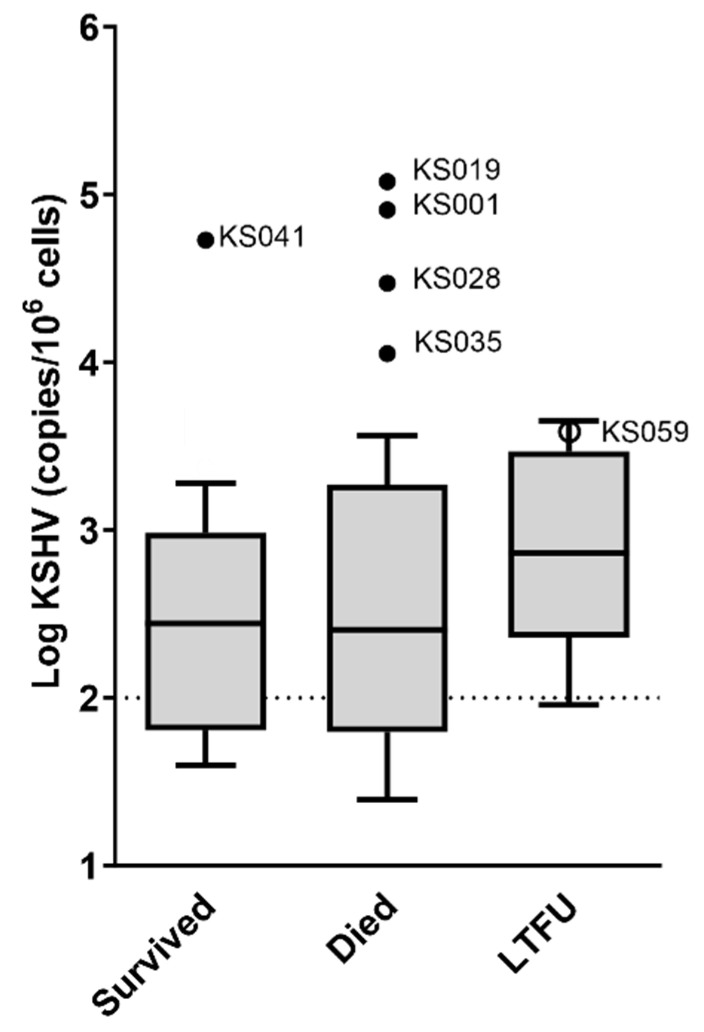
The association of KSHV VL to outcome among KS patients (*n* = 94). KSHV viral load outliers identified in Figure 2C are indicated with dots in these graphs (solid dots indicate outliers in the distributions here as well as in Figure 2C, whereas empty dots indicate that the values are not considered outliers in this distribution). The dotted lines indicate the cut-off (100 copies/10^6^ cells) for elevated KSHV viral load. Medians are compared statistically using the Kruskal–Wallis test; *p* values > 0.05 are not shown. Samples without detectable KSHV DNA are excluded.

**Table 1 viruses-16-00189-t001:** Demographic, clinical and virological characteristics of participants with KS (*n* = 94).

Variable		N (%) or Median (IQR)
Sex	Male	59 (63%)
	Female	35 (37%)
Age (years)		37 (32–42)
On ART	Yes	91 (97%)
	No	1 (1%)
	Unknown	2 (2%)
On ART for >1 year	Yes	8 (9%)
	No	30 (32%)
	Unknown	56 (59%)
Defaulted on ART	Yes	15 (16%)
	No	2 (2%)
	Unknown	77 (82%)
KS staging	T_0_	3 (3%)
	T_1_	91 (97%)
	Lymphedema	54 (59%)
	Extensive cutaneous disease	20 (22%)
	Pulmonary KS	12 (13%)
	GIT/another visceral site	6 (6%)
	I_0_	48 (51%)
	I_1_	46 (49%)
	S_0_	85 (90%)
	S_1_	9 (10%)
Response to treatment	Responded	40 (43%)
	Complete response	3 (3%)
	Partial response	37 (39%)
	Did not respond/progressed	12 (13%)
	Progressive disease	2 (2%)
	Relapse	10 (11%)
	Died	20 (21%)
	Unknown	22 (23%)
Survival at long-term follow-up	Survived	36 (38%)
	Died	36 (38%)
	LTFU	22 (24%)
CD4 count at diagnosis (cells/µL)		186 (55–341)
Haemoglobin (g/dL)	Females	10.1 (9.0–11.4)
	Males	11.2 (9.8–13.1)
White cell count (×10^9^ cells/L)		6.7 (5.0–8.8)
Platelets (×10^9^ cells/L)		320.5 (232.0–413.0)
Albumin (g/L)		34 (26–40)
hIL-6 (pg/mL)		9.9 (4.4–23.3)
HIV VL (copies/mL)		49 (1–5553)
KSHV VL	Not detectable	30 (32%)
	Detectable	64 (68%)
	Elevated >100 copies/10^6^ cells	46 (49%)
	KSHV VL (copies/106 cells)	280.5 (69.7–1727.3)

Sex, ART, ART for >1 year, ART default status, KS staging, response to treatment, survival and KSHV VL status are given as count and percentage; age, CD4 count, haemoglobin, white cell count, platelet count, albumin, hIL-6, HIV VL and KSHV VL are given as median and range. Normal ranges are as follows: haemoglobin for females 12–15 g/dL, males 13–17 g/dL; white cell count > 3.9 × 10^9^/L; platelets for females 186–454 × 10^9^/L, males 171–388 × 10^9^/L; albumin 35–52 g/L; hIL-6 < 1.8 pg/mL). Missing data are excluded listwise.

**Table 2 viruses-16-00189-t002:** Binomial logistic regression results for the association of KSHV viral load with S1 stage of KS (*n* = 94).

Characteristic	Unadjusted Odds Ratio	95% CI for Unadjusted Odds Ratio		Adjusted Odds Ratio	95% CI for Adjusted Odds Ratio		*p* Value
		Lower	Upper		Lower	Upper	
KSHV viral load ^1^	1.79	0.78	4.14	5.55	1.28	24.14	0.022 *
Treatment stage: ^2^							
Mid-chemotherapy	0.31	0.06	1.70	0.62	0.05	7.90	0.71
Post-chemotherapy	0.75	013	4.31	7.70	0.28	211.04	0.23
Sex ^3^	0.45	0.09	2.30	0.00	0.00	0.00	0.99
Age	0.96	0.88	1.05	0.95	0.81	1.11	0.50
CD4 count ^4^	1.00	1.00	1.00	1.01	1.00	1.01	0.070
Host interleukin 6 (IL-6)	1.01	0.99	1.02	1.00	0.98	1.02	0.94

^1^ KSHV viral load is log-transformed. ^2^ Treatment stage at sample collection reference is pre-chemotherapy. ^3^ Sex reference is male. ^4^ CD4 count at time of KS diagnosis. * Indicates a statistically significant variable at *p* < 0.05.

**Table 3 viruses-16-00189-t003:** Selected clinical and virological characteristics of a subset of patients with extremely high KSHV VL (*n* = 6) categorized as statistical outliers compared to the remainder of the cohort (*n* = 88).

	KSHV VL Outliers (*n* = 6)	Remainder of Cohort (*n* = 88)	*p* Value
Male sex	4 (67%)	55 (63%)	0.838
Age (years)	38.5 (35.7–44.5)	35.5 (31.3–41.0)	0.258
K8.1 OD	3.7 (2.6–3.7)	2.6 (2.4–3.7)	0.210
LANA OD	1.5 (0.6–2.7)	3.9 (0.9–7.2)	0.137
KS staging			
T_1_	6 (100%)	85 (97%)	0.646
Lymphedema	0 (0%)	54 (63%)	-
Extensive cutaneous disease	0 (0%)	20 (23%)	-
Pulmonary KS	4 (67%)	7 (8%)	<0.001 *
GIT/another visceral site	2 (33%)	5 (6%)	0.014 *
I_1_	6 (83%)	41 (47%)	0.107
S_1_	2 (33%)	7 (8%)	0.100
Response to treatment			
Complete response	0 (0%)	3 (3%)	
Partial response	2 (33%)	35 (40%)	
Progressive disease	0 (0%)	2 (2%)	0.687
Relapse	0 (0%)	10 (11%)	
Died	1 (17%)	19 (22%)	
Unknown	3 (50%)	19 (22%)	
Survival at long-term follow-up			
Survived	1 (17%)	35 (40%)	
Died	4 (66%)	32 (36%)	0.324
LTFU	1 (17%)	21 (24%)	
CD4 count at diagnosis (cells/µL)	57.0 (42.5–317)	202.5 (55.8–345.0)	0.484
HIV VL (copies/mL)	44.0 (1.0–3236.5)	49.0 (1.0–7348.3)	0.668
Anaemic	6 (100%)	65 (75.6%)	0.168
White cell count (×10^9^ cells/L)	7.7 (6.7–10.0)	6.4 (4.9–8.7)	0.209
Platelets (×10^9^ cells/L)	392 (252–518)	300 (229–403)	0.243
hIL-6 (pg/mL)	22.36 (11.77–66.10)	9.86 (4.40–23.05)	0.040 *

Sex, KS staging, response to treatment, survival and anaemia status are given as count and percentage; age, K8.1 OD, LANA OD, CD4 count, HIV VL, white cell count, platelet count and hIL-6 concentration are given as median and range. Normal ranges are as follows: white cell count > 3.9 × 10^9^/L; platelets for females 186–454 × 10^9^/L, males 171–388 × 10^9^/L; hIL-6 < 1.8 pg/mL). *p* values for categorical variables are determined by chi-squared or Fisher exact test as appropriate and are corrected for multiple comparison where appropriate, and for continuous variables they are determined by Mann–Whitney test. * indicates statistical significance at *p* < 0.05.

## Data Availability

The data that support the findings of this study are not openly available due to reasons of confidentiality (human data) and are available from the corresponding author, Z.M., upon reasonable request. The data are located on a secure RedCap repository.

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
