# Peer review of "Clinical Significance of Elevated KSHV Viral Load in HIV-Related Kaposi’s Sarcoma Patients in South Africa"

_viruses, 2024, doi:10.3390/v16020189_

Round 1

Reviewer 1 Report

Comments and Suggestions for Authors

This manuscript describes a retrospective cohort analysis in South African persons living with HIV that seeks to define whether KSHV viral load in peripheral blood has predictive value for outcomes of Kaposi sarcoma. Several variables are considered relative to KSHV loads, including biological sex, CD4 count, HIV load, ART status and adherence, anemia, and level of inflammatory cytokine IL-6. These are considered as possible covariates and correlates to tumor and immune status, as well as the presence of systemic disease. The analyses indicate that higher KSHV blood DNA detection strongly correlates with the presence of systemic HIV-related disease. It is also interesting that a handful of statistical outliers, designated as having “very high” KSHV levels, seem to have a very poor prognosis. The study is important and highly relevant to the field of KSHV epidemiology, as there is an unmet need for predictive criteria in KS disease prognosis and treatment. However, many of the data are quite frankly confounding. Based on the data presented, beyond the finding that KSHV viral load is an apparent risk factor for systemic HIV-related disease, it is difficult to determine whether the main conclusions of the study are justified by the results and analyses presented. Major and minor concerns are listed below.

Major concerns:

1. 49% of the patient cohort has CD4 counts less than 200/mm3, which is indicative of AIDS, with many more apparently near 200/mm3 based on the scatter plot in Fig. 1A and likely immune suppressed. This does not align with data stating that 99% of the cohort has been on ART for greater than 1 year. HIV loads were available but were not integrated into analyses or conclusions. One wonders whether there is a linear relationship between KSHV loads and low CD4 counts or high (or detectable) HIV loads. These analyses were not provided.

2. The clinical disease criteria are poorly defined. They are essentially not defined in the study participant section of the manuscript, and then only get brief mentions in various sections of the paper. This limits the readers ability to comprehend interpretations and put them into clinical context. For instance, why does KSHV viral load correlate with systemic HIV-related disease (S1), yet the majority of persons defined as having “very high” KSHV loads fall into the S0 category. Further complicating matters, 4 of these persons died. All six statistical VL outliers are listed as I1 – severe immune suppression. Is the only criterion here a CD4 count below 200/mm3?

Minor concerns:

1. This statement needs reconsideration or clarification, as there are endemic forms of KS that do not require HIV as a comorbidity (lines 41-43). “The gammaherpesvirus, Kaposi’s Sarcoma-associated Herpesvirus (KSHV), is the causative agent of KS, but while necessary, is not sufficient for KS development, requiring immune suppression often due to HIV infection.” Indeed, KS+ but HIV- is listed as an exclusion criterion for your patient cohort.

2. This statement is hypothetical, but is written as fact (lines 43-45). Please reword. “Both HIV infection and KSHV lytic infection promote cytokine production which in turn initiate KS tumour development and progression.”

3. This statement (lines 65-66) implies that the link is causal, when it is actually correlative: “development of KS results from an interplay of HIV-related immune suppression and KSHV DNAemia.”

4. Line 327: Going back to the need to better define the staging criteria, what is meant by “B symptoms”?

Reviewer 2 Report

Comments and Suggestions for Authors

The authors sought to establish an association between the KSHV viral load in patients' blood and clinical outcomes by retrospectively analyzing a cohort of 94 KSHV-positive patients. While this reviewer acknowledges the effort and originality of the study, there are notable gaps in the data that are crucial to supporting the hypothesis, and there's a lack of experimental detail. 

Comments: 

1.     The statistical analysis serves as the cornerstone of this work. It would greatly benefit from a more elaborate explanation of the methods used for each comparison across all figures. Additionally, the absence of statistical significance in the figures is a notable concern. 

2.     Including the normal range of different parameters, such as CD4 cell counts, would aid readers in interpreting the results more effectively. 

3.     Clarification is needed regarding whether the viral DNA level was measured from KSHV DNA in patient PBMCs or plasma. 

4.     The classification of different stages (T, I, S) appears somewhat arbitrary. Providing more details on the criteria used for determination would enhance clarity and understanding. 

5.     The Materials and Methods section appears overly verbose and could benefit from streamlining for better readability and clarity.

Round 2

Reviewer 1 Report

Comments and Suggestions for Authors

My concerns were addressed. The manuscript is much improved and points of confusion have been clarified.

Reviewer 2 Report

Comments and Suggestions for Authors

The authors provided point-to-point revision and more scientific soundness in the newer version of manuscript.